# Well-Being and the Lifestyle Habits of the Spanish Population: The Association between Subjective Well-Being and Eating Habits

**DOI:** 10.3390/ijerph18041553

**Published:** 2021-02-06

**Authors:** Laura Cabiedes-Miragaya, Cecilia Diaz-Mendez, Isabel García-Espejo

**Affiliations:** 1Research Group in Sociology of Food, Department of Applied Economics, University of Oviedo, 33006 Oviedo, Spain; lcabie@uniovi.es; 2Research Group in Sociology of Food, Department of Sociology, University of Oviedo, 33006 Oviedo, Spain; igarcia@uniovi.es

**Keywords:** subjective well-being, life satisfaction, eating habits, Mediterranean diet

## Abstract

The so-called Mediterranean diet is not simply a collection of foodstuffs but an expression of the culture of the countries of the south of Europe, declared Intangible Cultural Heritage by UNESCO. Despite the link between food and culture, little has been studied about how diet contributes to the well-being of the population. This article aims to analyze the association between subjective well-being and the eating habits of the Spanish population in order to gain a better understanding of the subjective well-being that food culture produces. For this study, we used a representative sample of the Spanish adult population from a survey by the Sociological Research Center (CIS 2017). Three indicators of subjective well-being were used: perceived health, life satisfaction, and feeling of happiness. The independent variables relating to eating habits considered in the analysis were, among others, how often meat, fish, vegetables, fruit, and sweets were consumed; how the food was prepared; how often meals were eaten out at restaurants or cafés and how often they were eaten with family or friends. Other independent variables related to lifestyle habits were also included in the analysis, in particular, physical exercise and body mass index. We used ordinal logistic regressions and multiple linear regression models. Our findings coincide in large measure with those obtained in earlier studies where perceived health and income play a key role in evaluating subjective well-being. In turn, several variables related to lifestyle habits, such as consuming sweets and fruits, social interaction around meals, exercising, and body mass index, were also associated with subjective well-being.

## 1. Introduction

Along with modern society’s increasing interest in looking after the body and health, citizens have shown greater interest in taking up healthy lifestyles that include specifically healthy eating and regular physical activity. This is not a recent phenomenon, as the World Health Organization (WHO) has defined lifestyle since the 1980s as “… a general way of living based on the interplay between living conditions in the wide sense and individual patterns of behavior as determined by sociocultural factors and personal characteristics” [1]. In line with economic well-being in developed societies, mass consumer society offers the possibility of following lifestyles that are more hedonistic or open to pleasure, generally associated with greater degrees of subjective well-being [2,3].

In the area of health, the relationship between eating and well-being is obvious, as the close link between health and diet enables the population to move towards better health by controlling the food that is eaten. Social sciences have also corroborated the relationship between eating and social life, showing how food, beyond being mere sustenance, contributes to bonding social groups around a table. The social interaction linked to food has been shown to be, in the specific case of Spain, a reflection of a food culture that is widespread throughout the Spanish population [4,5].

The so-called Mediterranean diet is not simply a collection of foodstuffs but an expression of the culture of the countries of the south of Europe, declared Intangible Cultural Heritage by UNESCO. However, the factors that make food a source of satisfaction are not known. Cultural studies about food over decades have clearly shown eating’s social aspects and connections and how it goes beyond mere sustenance, as the patterns of social behavior behind meals give value to the food [6]. The sociability intrinsic to eating practices, the meanings that are attributed to their selection and rejection, and the inevitable connection between food and its visible effects on the body, demonstrate that food involves perceptions, values, and meanings that make eating a social act and not simply biological behavior [7,8]. 

The objective of this article is to analyze the association between subjective well-being and the eating habits of the Spanish population, in order to gain a better understanding of the subjective well-being that food culture produces. 

Although the terms are sometimes considered interchangeable, most scholars differentiate happiness, satisfaction with life, and subjective well-being. In general, the concept of life-satisfaction is associated with the cognitive component of subjective well-being, referring to a person’s overall evaluation of their life (“life in general”), based on a process of conscious reflection. It involves the individual’s holistic perception and is related to the perceived distance between expectations and achievement. In contrast, the concept of happiness is associated with the affective component of well-being and is related to the presence of positive feelings at a given moment [9,10,11,12,13,14,15,16,17,18,19]. Meanwhile, self-assessed health is not always directly related to the actual state of health [16,20], but is regarded as hugely important as a measure of a population’s general health [21,22,23].

The literature available on the factors related to subjective well-being is plentiful, and includes factors associated with lifestyle to an ever greater extent. The contributions by economists have started increasing intensively since the 1990s, around the so-called economics of happiness [12,13,24], and studies looking for the objective benefits of subjective well-being have been developed [25]. Subjective well-being has been positively related to important and diverse societal domains such as productivity, health, longevity or creativity, and is starting to contribute to well-being being placed in a more center-stage position in policy making [25]. However, studies linking subjective well-being and food have not acquired special relevance until recent years, despite its potential impact in terms of societal benefits so important as increasing longevity, improving productivity or reducing healthcare costs. 

As Veenhoven has already established in his review of nutrition and happiness using the *World Database of Happiness* [26], research that includes food is based on surveys designed for specific purposes and/or is limited to a specific segment of the population (adolescents, university students, the elderly).

Reviewing the Spanish literature, furthermore, allows us to confirm that most articles are in journals dedicated to health and, particularly, food consumption [27,28,29,30,31]. At the same time, the National Health Survey (by the National Institute for Statistics, INE) [32] does not include variables that allow for a social view of eating habits or complementary information to associate frequency of consumption with well-being. This study aims to remedy these faults with data from a national survey representing the Spanish population to gain an understanding of the relation between subjective well-being and eating habits. The main hypothesis guiding this study is that there is a relationship between the Spanish population’s eating habits and their subjective well-being, or in other words, that food can generate a feeling of well-being. Self-assessed health, life-satisfaction, and self-reported happiness are associated with eating habits.

## 2. Materials and Methods

The database used is the CIS’s Poll 3170 from March 2017 [33]. The CIS (Sociological Research Center) is a technical service of the general state administration in Spain, in charge of the scientific study of the Spanish society. The CIS applies scientific methods in order to obtain representative samples of the Spanish population. The participation in the poll is voluntary. Once the volunteers have accepted to participate, they can decide not to answer some questions. However, all the answers included in the poll are anonymous and protected by the laws of statistical secrecy and data protection. Besides, no personal data or identifiers of the person who has responded are kept (these characteristics of the poll and the correspondent questionnaire can be seen with more detail at http://www.cis.es/cis/opencms/ES/1_encuestas/ComoSeHacen/pasosencuesta.html#pasos2 (accessed on 25 January 2021) and http://www.cis.es/cis/export/sites/default/-Archivos/Marginales/3160_3179/3170/cues3170.pdf (accessed on 25 January 2021)).

The total sample interviewed was 2487 people of Spanish nationality, aged 18 or over. Data were analyzed using IBM SPSS Statistics 24 (IBM, Armonk, NY, USA).

Dependent variables included self-rated assessment of the state of health (question 22), the degree of satisfaction with life in general (question 9), and the assessment of perceived happiness or unhappiness (question 25). The state of health is rated on a scale of 1 to 5 with 1 being “very good”, 2 “good”, 3 “fair”, 4 “bad”, and 5 “very bad”. The degree of satisfaction with life in general is on a scale from 0 to 10, where 0 is “completely unsatisfied” and 10 is “completely satisfied”. The degree of perceived happiness is also on a scale from 0 to 10, with 0 “completely unhappy” and 10 “completely happy”.

With the self-assessed state of health, we carried out an ordinal logistic regression in which the values of the dependent variable were grouped into three values, given that the percentage of people who rated their health as “bad” or “very bad” is negligible. Thus, the value 1 refers to “bad or very bad rating”, the value 2 to “fair”, and the value 3 to “good or very good”. With the dependent variables for life-satisfaction and happiness rating, we modelled multiple linear regressions. Since we considered that the relation between these three variables might be bidirectional, the assessment of state of health was included as an independent variable within the other two models.

Independent variables included basically sociodemographic variables and variables relating to lifestyle habits, mostly eating habits. The sociodemographic variables included were the following: sex, age, size of community, marital status, type of household, education, working situation, and household income. As variables relating to eating habits, we considered how often meat, fish, vegetables, fruit, and sweets were consumed; how the food was prepared (personally cooked or ready-made meals); how often meals were eaten out at restaurants or cafés; how often with family or friends; how often food was taken from home or bought to be eaten outside the home. Last, we also included body mass index (BMI), how often exercise was taken, and the nature of the personal economic situation as independent variables. The BMI was calculated, following the WHO criteria, as a person’s weight in kilograms divided by the square of the height in meters. For most adults, 18·5–24·9 kg/m^2^ is considered a normal BMI range: within the healthy weight range.

Table 1 shows the sociodemographic characteristics of the population studied, according to sex (to access the cross-over of other variables, please see the link available at [33]). Out of the 2487 participants in the CIS’s Poll 3170, 1271 were women (51.1%) and 1216 (48.9%) were men. The mean age was 50 years (51 the women and 49 the men). More than a half of the participants were married (50.7% of the women and 53.3% of the men). Almost two thirds of the households consisted of couples with or without children, (to a greater extent with children). With respect to the educational level, the weight represented by women in both extremes is higher than the represented by men. This pattern is repeated in relation to the self-assessed personal economic situation, though less markedly. The opposite pattern applies to household income, with a higher proportion of men reporting values of household income in the extremes.

Among the results of the analysis presented below, we only present the variables that showed some statistical significance in the regressions carried out. At the same time, in this study we did not aim to quantify probabilities but to analyze “correlation” or “association” of the variables that appeared significant, analyzing their effect with all other factors being equal. 

The main hypothesis of this study is to confirm the association between eating habits and subjective well-being. An additional hypothesis was considered that referred to the association between other lifestyle habits also related, though not so directly, with eating habits, and subjective well-being. These hypotheses are expressed through six questions: the main hypothesis is expressed in the first three questions and the additional hypothesis in the last three—Is the frequency of consumption of healthy produce (fruit/vegetables and fish) associated with greater subjective well-being? Does eating in company make us feel happier? Is eating ready-made food associated with a more negative assessment of health? Does feeling healthy contribute to greater degrees of subjective well-being? Are people who have obesity happier? Does exercise improve well-being? 

## 3. Results 

Before presenting the results obtained in the regressions carried out, let us show the aggregated values of the dependent variables considered in the research (Table 2). Men reported good or very good perceived health in greater proportion than women (75.1% vs. 67.7%, respectively). The mean rating of the satisfaction with life reported was 7.30 (range 0–10) in both cases. However, the mean happiness rating reported by the men was slightly higher than the reported by women (7.74 vs. 7.67, range 0–10, respectively).

### 3.1. Assessment of State of Health 

Spanish people as a whole have a mainly positive assessment of their general state of health. Only 5.0% of Spaniards consider their health bad or very bad, 58.3% consider their health good, and 13.0% perceive their health as very good. As can be seen in Table 3, the sociodemographic variables most commonly associated with a good state of health are a person’s sex and either living as a couple with children or living with mother and/or father (with or without siblings). Men generally give a more positive assessment than women, particularly before the age of 30. Some variables related with eating habits also produce significant results, in particular the social relations surrounding meals. Thus, having lunch or dinner with family or friends several times a month appears to be a factor that affects how the state of health is rated. Interestingly, eating sweets several times a week also produces positive results. 

Rating one’s personal economic situation as good has a great influence on self-assessed health (curiously, to a greater extent than rating it as very good), as is engaging in physical exercise and having a normal body mass index. A high level of happiness is also associated positively with a high level of perceived health (Table 3).

### 3.2. Satisfaction with Life

In general, Spanish people feel fairly satisfied with their life, with a score of 7.3 out of 10. As shown in Table 4, age and household income are the only two sociodemographic variables that emerge as being associated positively with a high level of life-satisfaction. With respect to household income, the people who are least satisfied with their life are those whose income falls below €1801, and very markedly when it is under €600 a month. People under 40 are most satisfied with their life, while, in relation to income, people who rate their personal economic situation as good or very good are most satisfied. Rating perceived health as good or very good is associated positively with a high level of satisfaction with life. 

Variables related to eating habits do not greatly affect general life-satisfaction, but it could be said that eating meat frequently is associated negatively with life-satisfaction, while frequent consumption of fruit is more favorable, pointing out a possible positive association of the Mediterranean diet with life-satisfaction (Table 4).

### 3.3. Personal Happiness

People in Spain rate their state of happiness positively, as the average score for the population is 7.7 out of 10. What factors affect happiness most significantly? It emerges that the most important variables are having a personal economic situation rated good or very good and having a state of health perceived as good or very good. Age is also a significant element: the younger a person is, the happier, especially between the ages 18 and 30. In contrast, the smaller the community in which a person lives, the lower their state of happiness will be, and having a university-level education or a low household income were found to have similar negative associations (Table 5).

The variables related to eating habits are unimportant in connection with feeling happy or unhappy. There is only a small positive association with happiness and the consumption of fish and a negative one with the consumption of meat. Engaging in physical exercise is even less important (Table 5).

## 4. Discussion

As the results presented above show, income matters and matters significantly. There is a vast array of literature on the relationship between money and happiness. These studies generally point to the existence of a positive effect from higher income but with diminishing returns and even, marginally, the possibility of an inverse causality [13,15,24,34]. The results obtained in our study are in line with previous research, and specifically in the context of Spain [35,36,37]. In turn, several variables related to eating habits, were also associated with subjective well-being. To place these findings in context, we compare the results presented above with the available literature. The objective is to confirm, in a practical and accurate manner, the relation between food and subjective well-being and to account for how this relation manifests specifically in the case of Spain.

### 4.1. Is the Frequency of Consumption of Healthy Produce Associated with Greater Subjective Well-Being? 

In practically all research reviewed, it can be said that, out of all the food groups, consumption of fruit and vegetables shows a strong and significant positive effect with respect to subjective well-being [26]. The literature examining the Spanish case in this area is sparse and found in publications on health matters, in cross-sectional studies focusing on the relation between general patterns of adherence to the Mediterranean diet and well-being. For example, Grao-Cruces et al. [28], using a sample of 1973 Andalusian teenagers (ages 11–18), conclude that the individuals who keep more closely to the Mediterranean diet are physically more active and show a higher level of satisfaction with life than those with a low-quality diet. However, as Blázquez et al. [29] demonstrate, the relationship between eating habits and self-assessed health is disputed. Specifically, in their study of people over 50, there was a very weak correlation (though statistically significant), between the number of aspects of healthy eating that individuals met and their self-assessed score on health. At the same time, by means of adjusted multiple linear regression models, Zaragoza et al. [30], basing their work on a sample of 351 people over 60, found a direct relationship between following a Mediterranean diet and life-satisfaction for women, though the association was not significant in the case of men. Lastly, Ferrer-Cascales et al. [31], using a sample of 527 teenagers, found a positive association between adherence to a Mediterranean diet and happiness.

In our study, life-satisfaction was only associated positively with consumption of fresh fruit. The association of other food groups with happiness is not significant. Curiously, those who ate sweets several times a week had a higher self-assessment for health than those who stated that they seldom or never ate them. The association found here could be related to the fact that those who enjoy a good basic state of health allow themselves this habit. 

### 4.2. Does Eating in Company Make Us Feel Happier? 

There are some previous studies in the academic literature that positively relate sharing meals with others and well-being. For example, in a non-Western setting, Yiengprugsawan et al. [38], using data from a cohort of 39,820 people in Thailand, found a correlation between frequently eating alone and both poor perceived health and unhappiness, concluding that sharing meals could contribute to improving people’s happiness. A positive relationship between the frequency of having dinner in company and happiness is shown by a sample of 400 elderly individuals (aged from 60 to 90) living in Central Chile, by means of ordered logit models [39]. 

In his survey of the literature on how much hedonism affects happiness, Veenhoven [2] underlines the positive relationship between leisure activities—especially outdoors, such as eating out and sport—and happiness. Even in a highly self-sufficient society in the Bolivian Amazon, social leisure activities show a positive association with happiness [40]. 

In our study, we cannot confirm that the fact of sharing meals is associated with greater happiness. There is, however, a very significant relationship between eating at the home of family and/or friends (especially if several times a month) and a higher level of self-rated health. Moreover, eating at a restaurant or similar establishment shows no significant relationship with any indicator of subjective well-being, though it must be borne in mind that the question in the survey we used mentions the location and does not ask if restaurants were visited alone or in company, nor whether visits were for work or for leisure reasons. These are important nuances that affect how far we can analyze the possibility of a positive interaction between company and eating.

### 4.3. Is Eating Ready-Made Food Associated with a More Negative Assessment of Health? 

We have found no previous studies on this area in the literature. In our case, we did not detect any significant association with any of the indicators of subjective well-being under consideration. It is true, however, that the category does not include any specific products and only divides meals into those prepared by oneself and those bought ready-made. 

### 4.4. Does Feeling Healthy Contribute to Greater Degrees of Subjective Well-Being? 

The positive association between subjective health and both life satisfaction and happiness, as shown in the empirical evidence available, is practically universal [36,41]. Our findings provide support for this assertion: those who rate their health as good or very good claim to be happier and show a higher level of satisfaction with life to a very significant degree. Our findings are in line with other studies carried out in Spain, including Videra-García and Reigal-Garrido [16] and Lera-López et al. [22,37]. 

### 4.5. Are People Who Have Obesity Happier? 

Despite the popular Spanish belief in the “happy fat person”, the academic literature on the subject suggests that obesity is rather associated with lower subjective well-being, which can be attributed to both its negative impact on health and also the social stigma associated with it [42]. While the results obtained in our study do not enable us to state that people with overweight feel happier, they do show an association between having a normal BMI and less life-satisfaction. This is an unexpected result, given that a BMI above the healthy range is associated in the literature with lower levels of subjective well-being, showing, for instance, the existence of a bi-directional linkage between obesity and depression [42].

### 4.6. Does Exercise Improve Well-Being?

The academic literature provides support for a positive relation between physical exercise and happiness (for example, in the systematic review carried out by Zhang and Chen [43]). This association is also found between exercising and life satisfaction (see for example Pettay [44], with almost 800 college students in the United States of America, and Lera-López et al. [22], with a sample of 816 individuals aged from 50 to 70 in Spain). Contrary to expectations, our analysis only found a positive relation between physical exercise and better self-rated health. This result agrees with that obtained by Blázquez et al. [29], with a sample of 781 people over 50 years old.

Baruth et al. [45], in a longitudinal study of 2132 initially sedentary men, concluded that men who were usually very happy and optimistic increased their physical activity significantly more than men who were not happy. In our case, we cannot tell if physical exercise has a positive influence on perceived health or whether those who perceived a good level of health tend to take exercise more often. However, those who exercise have a more positive assessment of their own health, especially those who do so once a week. 

## 5. Conclusions

The data that we present here enable us to explore the relation between subjective well-being and eating. Individuals’ assessments of their own state of health, their satisfaction with life, and their level of happiness allow us to understand how far food contributes to well-being. Corroborating this relation with statistically representative data about the Spanish population is something that is new in an area that is usually approached through ad hoc studies. Moreover, it is particularly important to confirm (or not) the relationship in a society such as Spain’s, where the Mediterranean diet forms part of the cultural identity and supports good eating habits. 

We have been able to confirm that being able to rate one’s personal economic situation favorably is particularly important and influences positively and significantly the self-assessed state of health, satisfaction with life, and level of happiness. At the same time, those who rate their state of health positively are the happiest and most satisfied with their lives, so that money and health are shown to be the keys to well-being. 

With regard to Spanish people’s well-being in relation to their eating habits, it seems that not only is the food they eat of particular importance, but also the people they eat with. As far as what is eaten is concerned, it is not possible to say that those who eat healthily show a higher level of subjective well-being, beyond suggesting a potential positive association between the Mediterranean diet and overall life satisfaction. All in all, it can be stated that the selection of foods per se is associated with well-being. This issue is especially relevant in the Spanish case, where a solid food culture shared by the population guides the choice of products that make up the Mediterranean diet. The data show greater degrees of well-being among those who eat more fresh fruit, as well as those who eat more sweets. These data could indicate that what really produces well-being is being able to choose. 

The data analyzed appear to indicate that subjective well-being is related to company (who one eats with), as, although the data show us “where one eats”, those who eat with family and friends have a better assessment of their health. Therefore, these surroundings make sociability the best context for a satisfying meal. The Spanish case is particularly paradigmatic with respect to food, since the family-centered environments, typical of Spanish culture, favor social relationships being played out around eating and therefore may enable a feeling of subjective well-being. 

We have further noted that there is a direct relation between engaging in more exercise and assessing one’s health positively, especially among those who take moderate exercise on a regular basis. We can assume that self-perceived health acts as a middle term between physical exercise and happiness; we cannot therefore say that someone who takes exercise is happier, but we can affirm that those who take exercise feel healthier. Thus, physical exercise results in a positive perception of the state of health that leads to subjective well-being. Essentially, the results show that we are not dealing with a direct relationship between food and well-being or exercise and well-being. 

However, we consider that there are at least two factors that are key to interpreting the subtleties of this relationship: on the one hand, sociability acts as a middle term between food and well-being; on the other hand, health is a middle term between physical exercise and well-being. The keys to this relationship lie in the way that individuals act to seek a source of satisfaction and well-being in food and in physical exercise.

The limitations of the present work include the fact that, as it deals with a cross-sectional study, apart from conjecturing some plausible interpretations, it does not permit us to draw conclusions about causality. 

The strong points that stand out from the research are its basis on a representative national sample from the population of Spain and with more variables than other Spanish works. This is something that is particularly important when we bear in mind that comparative studies on subjective well-being carried out at an international level have not included the Spanish case [26]. The fact that our study has considered variables that have seldom been examined in the academic literature on subjective well-being, including the situation of eating out or concern about the consumption of ready-made meals at home, is also important. In summary, this is the only study conducted in Spain with these points of focus, and we consider it a useful, though modest, contribution to the literature on eating and well-being.

We should not conclude this analysis without drawing attention to an element that underlies this research: the relevance and the limitations of Spanish surveys to capture the food cultural aspects. It is clear that we need to have reliable data to corroborate our hypotheses, however, some surveys, such as the one used here, are not fully capable to capture the sociability or the food meanings that explain the decisions about what and how to eat. This deficiency constitutes a problem for advancing the knowledge of the feeding habits in societies like Spain, in which a solid alimentary culture is central. With quantitative methodologies or qualitative approaches, we consider it convenient to continue researching about the positive social interaction around meals, in order to gain a deeper understanding about our Mediterranean diet. 

## Figures and Tables

**Table 1 ijerph-18-01553-t001:** Sociodemographic characteristics of the study population.

	Total (*n* = 2487)	Women (*n* = 1271)	Men (*n* = 1216)
**Mean Age (Years)**	49.99	50.99	48.94
**Marital status (%)**			
Married	52.0	50.7	53.3
Single	32.3	28.2	36.5
Widowed	8.0	13.5	2.4
Separated	2.7	2.8	2.7
Divorced	4.7	4.5	4.9
**Type of household (%)**			
Single	12.7	13.2	12.2
Single with children	6.6	10.3	2.6
Couple with children	37.9	38.5	37.3
Couple without children	25.1	23.6	26.7
Living with mother and/or father	14.6	11.4	17.8
Other	2.8	2.7	3.0
**Educational level (%)**			
Primary or no education	23.2	26.2	20.2
Secondary (first cycle)	24.3	21.8	26.9
Secondary (second cycle)	14.2	13.1	15.3
Vocational training	17.2	16.3	18.1
University	20.9	22.5	19.2
**Self-assessed personal economic situation (%)**			
Very Good	1.5	1.6	1.4
Good	35.9	34.8	37.2
Fair	46.8	47.5	46.0
Bad	11.7	11.6	11.8
Very bad	3.8	4.2	3.4
**Household income (EUR/month, %)**			
EUR 600 or less	6.2	5.8	6.5
Between EUR 601 and EUR 900	10.9	12.3	9.4
Between EUR 901 and EUR 1200	12.1	12.1	12.0
Between EUR 1201 and EUR 1800	15.1	13.7	16.5
Between EUR 1801 and EUR 2400	9.8	9.3	10.4
Between EUR 2401 and EUR 3000	6.8	6.7	7.0
More than EUR 3000	5.6	5.3	5.7

Source: CIS. Estudio 3170 (2017).

**Table 2 ijerph-18-01553-t002:** Dependent variables: Satisfaction with life, perceived health and happiness.

	Total (*n* = 2487)	Women (*n* = 1271)	Men (*n* = 1216)
**Satisfaction with life (Mean, 0–10 scale)**	7.30 (1.84) *	7.30 (1.90) *	7.30 (1.77) *
**Perceived health (%)**			
Good or very good	71.3	67.7	75.1
Fair	23.6	26.0	21.1
Bad or very bad	5.0	6.2	3.8
**Happiness (Mean, 0–10 scale)**	7.71 (1.66) *	7.67 (1.74) *	7.74 (1.56) *

* SD: Standard deviation. Source: CIS. Estudio 3170 (2017).

**Table 3 ijerph-18-01553-t003:** Ordinal logistic regression. Assessed state of health.

	Rating
**Sex** (Base: woman)	0.439 ***
**Age** (Base: over 70)	
Between 18 and 30	0.933 **
Between 31 and 40	0.502
Between 41 and 50	−0.154
Between 51 and 60	−0.432
Between 61 and 70	0.655
**Marital status** (Base: divorced)	
Married	−0.587 *
Single	−0.155
Widowed	−0.213
Separated	0.264
**Type of household** (Base: other)	
Single	0.380
Single with children	0.760
Couple with children	0.937 **
Couple without children	0.483
Father mother with or without siblings or relatives	0.955 **
**Self-assessed personal economic situation** (Base: very bad)	
Very good	2.943 **
Good	1.229 ***
Fair	0.582 *
Bad	0.263
**BMI** (Base: obese)	
Normal	0.788 ***
Overweight	0.525 ***
**Frequency of physical exercise** (Base: unable)	
Daily	1.940 ***
Several times a week	1.802 ***
Once a week	2.327 ***
Several times a month	1.501 ***
Never or almost never	1.724 ***
**Sweet consumption** (Base: never or almost never)	
Daily	0.245
Several times a week	0.718 ***
Once a week	0.244
Less than once a week	0.092
**Lunch/dinner at the home of family or friends** (Base: never)	
Almost every day	−0.005
Several times a week	0.440
Several times a month	0.621 ***
Once a month	0.553 **
Several times a year	0.543 **
Once a year	0.405
**Scale of life-satisfaction (scale from 0 to 10)**	0.072 *
**Scale of happiness (scale from 0 to 10)**	0.295 ***
***N*:** 2478	

* *p* < 0.100; ** *p* < 0.050; *** *p* < 0.010.

**Table 4 ijerph-18-01553-t004:** Multiple linear regression. Scale of general satisfaction with life (0–10).

	B
**Age** (Base: over 70)	
Between 18 and 30	0.792 (0.274) ***
Between 31 and 40	0.256 (0.124) **
Between 41 and 50	0.105 (0.079)
Between 51 and 60	−0.007 (0.053)
Between 61 and 70	−0.056 (0.034)
**Household income** (Base: over EUR 3000/month)	
EUR 600 or less	−0.706 (0.260) ***
Between EUR 601 and EUR 900	−0.236 (0.115) **
Between EUR 901 and EUR 1200	−0.142 (0.070) **
Between EUR 1201 and EUR 1800	−0.105 (0.048) **
Between EUR 1801 and EUR 2400	−0.046 (0.039)
Between EUR 2401 and EUR 3000	−0.041 (0.034)
**Assessment of personal economic situation** (Base: very bad)	
Very good	1.961 (0.432) ***
Good	0.861 (0.120) ***
Fair	0.487 (0.074) ***
Bad	0.140 (0.059) **
**BMI** (Base: obese)	
Normal	−0.272 (0.120) **
Overweight	−0.091 (0.060)
**Meat consumption** (Base: never or almost never)	
Daily	−0.482 (0.333)
Several times a week	−0.263 (0.157) *
Once a week	−0.170 (0.109)
Less than once a week	−0.162 (0.109)
**Fruit consumption** (Base: never or almost never)	
Daily	0.611 (0.235) **
Several times a week	0.411 (0.241) *
Once a week	0.057 (0.282) **
Less than once a week	0.548 (0.329) *
**Assessment of state of health** (Base: good or very good)	
Bad or very bad	−1.345 (0.210) ***
Fair	−0.448 (0.109) ***
***N*:** 2478	

B: regression coefficients. Values in parentheses are standard error * *p* < 0.100; ** *p* < 0.050; *** *p* < 0.010.

**Table 5 ijerph-18-01553-t005:** Multiple linear regression. Scale for happiness (0–10).

	B
**Age** (Base: over 70)	
Between 18 and 30	0.753 (0.243) ***
Between 31 and 40	0.240 (0.110) **
Between 41 and 50	0.151 ((0.070) **
Between 51 and 60	0.041 (0.047)
Between 61 and 70	0.025 (0.030)
**Size of community** (Base: over 1,000,000 inhabitant)	
Less than or equal to 2000 inhabitants	−0.640 (0.193) ***
Between 2001 and 10,000 inhabitants	−0.177 (0.078) **
Between 10,001 and 50,000 inhabitants	−0.085 (0.049) *
Between 50,001 and 100,000 inhabitants	−0.109 (0.041) ***
Between 100,001 and 400,000 inhabitants	−0.047 (0.030)
Between 400,001 and 1,000,000 inhabitants	−0.046 (0.030)
**Marital status** (Base: divorced)	
Married	−0.085 (0.205)
Single	0.015 (0.099)
Widowed	−0.131 (0.079) *
Separated	0.095 (0.065)
**Educational level** (Base: primary or no education)	
University	−0.437 (0.157) ***
Secondary (second cycle)	−0.117 (0.053)
Secondary (first cycle)	−0.042 (0.032)
Vocational training	−0.101 (0.073)
**Household income** (Base: over EUR 3000/month)	
EUR 600 or less	−0.249 (0.232)
Between EUR 601 and EUR 900	−0.122 (0.102)
Between EUR 901 and EUR 1200	−0.124 (0.062) **
Between EUR 1201 and EUR 1800	−0.032 (0.043)
Between EUR 1801 and EUR 2400	−0.026 (0.035)
Between EUR 2401 and EUR 3000	−0.015 (0.030)
**Assessment of personal economic situation** (Base: very bad)	
Very good	1.778 (0.385) ***
Good	0.712 (0.107) ***
Fair	0.376 (0.066) ***
Bad	0.195 (0.053) ***
**Frequency of physical exercise** (Base: unable)	
Daily	0.104 (0.257)
Several times a week	−0.126 (0.132)
Once a week	−0.079 (0.097)
Several times a month	−0.023 (0.075)
Never or almost never	−0.085 (0.051) *
**Meat consumption** (Base: never or almost never)	
Daily	−0.364 (0.297)
Several times a week	−0.208 (0.140)
Once a week	−0.174 (0.097) *
Less than once a week	−0.161 (0.089) *
**Fish consumption** (Base: never or almost never)	
Daily	0.478 (0.265)*
Several times a week	0.097 (0.094)
Once a week	0.020 (0.064)
Less than once a week	0.042 (0.060)
**Assessment of state of health** (Base: good or very good)	
Bad or very bad	−1.517 (0.188) ***
Fair	−0.601 (0.097) ***
***N*:** 2478	

B: regression coefficients. Values in parentheses are standard error. * *p* < 0.100; ** *p* < 0.050; *** *p* < 0.010.

## Data Availability

The data is owned by the Centro de Investigaciones Sociológicas (CIS), Government of Spain.

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
