# Peer review of "Well-Being and the Lifestyle Habits of the Spanish Population: The Association between Subjective Well-Being and Eating Habits"

_ijerph, 2021, doi:10.3390/ijerph18041553_

Round 1

Reviewer 1 Report

This is an interesting piece of work on wellbeing and eating habits of people in Spain and how certain variables relate to wellbeing. There is a clear rationale for this research and the abstract is well written. The first part of the introduction is good but the second part (lines detailed below) doesn't seem to fit. I would like to see a clearer explanation of the methods, for example software used. Some of the language used is quite casual and could be made more scientific.

Line 29 - start with a full definition of WHO before abbreviating.

Line 74/75 - missing word

I feel that the introduction is a little long, with lots of information in lines 56-87 that doesn't sit well in the introduction. Perhaps consider if all of this is required.

Line 106 - perhaps a different word to "tiny" as I didn't think the word percentage would be coming after it. 

Could more details about significant results e.g. values be included in line 141 for example. 

Line 186 - not sure repeat of matters is necessary here. Sounds very casual.

Perhaps start the discussion with a very brief overview of what "the results presented above" show? Just for clarity and flow between the sections. 

Line 205 - the statement "practically all research" seems a little flippant

Line 270 - what is "normal BMI"? Within the "healthy" range? Can you elaborate any more on the unexpected result?

Line 274 - found out?

Line 309 - "probably" is a little casual

Line 311 - use of "everything", do you mean all data?

Lines 350-352 - check for clarity - "understanding"

Author Response

Reviewer 1.

We are very grateful for all the detailed remarks and very precise suggestions that you have made to this research.

Point 1.Languaje and introductin

All of them have been reflected in the new version of the text (including the elimination of several lines of the introduction and at the end of the results section).

Point 2.Methods

For instance, with respect to the methods, we had missed relevant aspects that we clarify now in the text; we have improved the flow between the sections; and we have explained what is considered “normal BMI”, as well as why we calified one of the results as “unexpected”.

In particular, in correlative order, your remarks have been considered in the following lines of the new version of the text: 8; 72-77; 86-87; 102-105; 162; 165-166; 177-183; 214-220; 227; 293-296; 300; 335; 337; and 378.

Reviewer 2 Report

The study's aim was to analyze the association between subjective well-being and the eating habits of the Spanish population in order to gain a better understanding of the subjective well-being that food culture produces. The subject of the paper is very interesting. The study presented indicators of subjective well-being (self-rated health status, life satisfaction and feeling of happiness) and their association with socio-demographic and lifestyle determinants, including food consumption. However, the research was not strictly focused on eating habits, e.g. the amount of food consumed was not taken into account (frequency consumption only), and some key pro-health products, such whole grains and nuts, were omitted. Moreover, according to the authors, eating habits include “physical exercise” and “body mass index” (abstract, line 18), but it is not the case. The fact that the study also extensively analyzed socio-demographic factors such as household income, economic situation, size of community, education level, etc.  leads me to strongly recommend changing the title of the manuscript from “eating habits” to “lifestyle habits” or “socio-demographic & lifestyle habits”.

The abstract needs some improvement as there is too much background and too few research results.

The method section (the study design and data collection) lacked key information, such as:

  • whether the approval of an ethical committee was obtained (name of committee, date and approval number)
  • if respondents gave their consent to the survey
  • study exclusion/inclusion criteria
  • description of questionnaire form, e.g. was the questionnaire validated?

At the beginning of the Results section, the overall characteristic of the population studied should be added in the form of a table and briefly described. To better understand what variables were used in the models, it would be helpful to show those results not mentioned because they were considered statistically insignificant (perhaps as supplementary material).  

Author Response

Reviewer 2.

We are very grateful for all the relevant recommendations you have made to this research. Virtually all of them have been reflected in the new version of the text. In particular:

Point 1. Title

We have changed the title of the manuscript in order to reflect the scope of the issue studied in the research.

Point 2. Abstract

The abstract was modified in order to be more coherent both with the text and the new title of the manuscript. 

Point 3. Methods, research design.

With respect of the data collection and the poll used in this research, we had missed relevant aspects that we clarify now in the text (lines 72-77). Besides, we have provided the link to this oficial poll (reference 33), where the methodological and ethical aspects are explained in more detail.

The overall characteristics of the population studied are presented in the new table 1, and they are briefly described in the text (lines 106-116).

Point 4. Results

The independent variables are enumerated in the text (lines 93-102) and given that we carried out three different regressions, due to the limitation of space we opted for only presented in the tables the variables that showed some statistical significance, as indicated in lines 123-124.

Reviewer 3 Report

This paper is clear and well written.  Methods are clearly described and seem appropriate to the study aims.  However, I find the level of originality/novelty and significance of this paper to be quite low. The benefits of a Mediterranean diet have been well-described in the literature for a long time. And there is a vast literature on well-being.

It is not clear why well-being is important. I think the paper would be strengthened by establishing the potential societal benefit to well-being, such as improved productivity, increased longevity, reduced economic burden of unemployment, lower healthcare costs, etc.

Comparisons to other countries would of course strengthen the assessment but I realize that is outside the scope of this paper.

I would encourage the authors to make a stronger case for the societal level benefit of well-being and why it would be important to achieve.  If a Mediterranean diet turns out to be closely associated with well-being (and not confounded by economic status) I think the results of their study would be more useful. 

Author Response

Reviewer 3.

We are very grateful for the relevant remarks that you have made to this research.

Point 1. Background and introduction

With respect to the remark that it is not clear why well-being is important, we have addressed this so relevant topic in the introduction, supplementing the academic literature with the information included in systematic reviews that have addressed it, because, as you say, this is a broad field of study (lines 46-55) .

Point 2 Results and conclusions.

We have stressed several conclusions of studies looking for the objective benefits of subjective well-being, in order to explain in short why subjective well-being is considered relevant both in terms of societal benefits and their potential significance in terms of the policy agenda. In this vein, we have stressed as well the relevance of subjective well-being in relation to diet, given its potential impact in terms of societal benefits so important as increasing longevity, improving productivity, or reducing healthcare costs.  We have reviewed the background related to the Mediterranean diet to precise the link with social and economic factors (lines 46-55)

Reviewer 4 Report

Need to edit abstract, stress the objective of this article and main hypothesis.

17-18 Factors related to food consumption are included as variables related to eating habits: sociability, physical exercise, and body mass index.

In this phrase, it is not clear which variables are dependent and which are independent.

Possibly «body mass index» is a dependent variable as well as a subjective well-being, and sociability, physical exercise are independent. Or… here they are intermediate variables?

119-120. “In the analysis, we also included body mass index (BMI), how often exercise was taken, and how the personal economic situation was assessed”. Are they intermediate variables?

91-93The main hypothesis should specified. What kind of relation? Formulate an alternative hypothesis.

125-132These six questions could combined with the main and additional hypothesis.

For 5th and 6 questions you need to formulate another hypothesis.

295-299These findings are interesting, but deviate from the main subject of the article.

300-309.Probably, on the contrary, level of subjective well-being, which, for example, depends on satisfaction with relationships with loved ones, including sexual relationships, determines the choice of products.

The analysis is interesting. I would like to wish the authors success in further studying the most important factors of subjective well-being of men and women.

Author Response

Reviewer 4.

We are very grateful for the relevant remarks that you have made to this research. Virtually all of them have been reflected in the new version of the text. In particular:

Point 1. Title, Abstract, and introduction

The abstract has been modified in order to clarify the character of the variables included in the study.

The objective of this article, apart from being indicated in the introduction, it is now clarified from the beginning, in the new title of the article.

Point 2. Methods

The dependent and independent variables are clearly identified in the new text (lines 78-80 and 93-102, respectively). Besides, in the new table 2 we present the aggregated values of the dependent variables considered in the research and they are briefly analyzed in the text, for women and for men separately (lines 139-145).

The main hypothesis of the research has been placed before the results section (lines 128-129). This serves to connect the objective of the study with the research questions. We have specified an additional hypothesis. The six questions are now combined with the main and additional hypothesis formulated (129-133).

Point 3. Analysis

With respect to the causality direction of the associations, we had indicated in the conclusions that, given that this study consists of cross-sectional analysis, apart from conjecturing some plausible interpretations, it does not permit us to draw conclusions about causality. Even so, in several parts of the text, we have suggested the possibility of a bidirectional causality (for instance, in relation to the choice of eating sweets and the assessment of health; lines 246-250).

Round 2

Reviewer 2 Report

The manuscript has been significantly improved. However, the methods (the study design and data collection) were not described in sufficient detail:

  • whether the approval of an ethical committee was obtained (name of committee, date and approval number)
  • if respondents gave their consent to the survey
  • study exclusion/inclusion criteria
  • description of questionnaire form, e.g. was the questionnaire validated?

Author Response

The survey used in this paper is included in the National Statistical Plan 2017-2020 and follows the recruitment and data protection procedures of the rest of the government statistics (Law 12/89, of May 9, of the Public Statistical Function)

The collaboration of the people selected to participate in a CIS survey is voluntary, but essential to ensure that its results reflect the opinions of society as a whole. At any time during the interview, the interviewee can choose not to answer any question if they so wish. All responses are anonymous, protected by the laws of statistical secrecy and data protection. 

No personal data or identifier of the person who responded is kept, and once the information they contain is recorded, the questionnaires are destroyed.

This information can be corroborated in the link to the questionnaire: http://www.cis.es/cis/export/sites/default/-Archivos/Marginales/3160_3179/3170/cues3170.pdf

We have summarized this information in line 73-84 (in red)

Reviewer 3 Report

I thank the authors for their revisions to this paper, which I think has improved its quality and utility in the peer reviewed literature.  I do think it would help if the paper was reviewed by an English language editor for minor improvements that could be made.  I recommend this paper be accepted with a final review for English language usage, if that is feasible. 

Author Response

The correct use of the English language. will be reviewed by the editorial team.

This manuscript is a resubmission of an earlier submission. The following is a list of the peer review reports and author responses from that submission.